# TLRs: Innate Immune Sentries against SARS-CoV-2 Infection

**DOI:** 10.3390/ijms24098065

**Published:** 2023-04-29

**Authors:** Stefania Mantovani, Barbara Oliviero, Stefania Varchetta, Alessandra Renieri, Mario U. Mondelli

**Affiliations:** 1Department of Research, Division of Clinical Immunology—Infectious Diseases, Fondazione IRCCS Policlinico San Matteo, 27100 Pavia, Italy; b.oliviero@smatteo.pv.it (B.O.); s.varchetta@smatteo.pv.it (S.V.); 2Medical Genetics, University of Siena, 53100 Siena, Italy; alessandra.renieri@unisi.it; 3Med Biotech Hub and Competence Center, Department of Medical Biotechnologies, University of Siena, 53100 Siena, Italy; 4Genetica Medica, Azienda Ospedaliero-Universitaria Senese, 53100 Siena, Italy; 5Department of Internal Medicine and Therapeutics, University of Pavia, 27100 Pavia, Italy

**Keywords:** Toll-like receptor, COVID-19, SARS-CoV-2, innate immunity, host genetics

## Abstract

Coronavirus disease 2019 (COVID-19), caused by severe acute respiratory syndrome coronavirus 2 (SARS-CoV-2), has been responsible for a devastating pandemic since March 2020. Toll-like receptors (TLRs), crucial components in the initiation of innate immune responses to different pathogens, trigger the downstream production of pro-inflammatory cytokines, interferons, and other mediators. It has been demonstrated that they contribute to the dysregulated immune response observed in patients with severe COVID-19. TLR2, TLR3, TLR4 and TLR7 have been associated with COVID-19 severity. Here, we review the role of TLRs in the etiology and pathogenesis of COVID-19, including TLR7 and TLR3 rare variants, the L412F polymorphism in TLR3 that negatively regulates anti-SARS-CoV-2 immune responses, the TLR3-related cellular senescence, the interaction of TLR2 and TLR4 with SARS-CoV-2 proteins and implication of TLR2 in NET formation by SARS-CoV-2. The activation of TLRs contributes to viral clearance and disease resolution. However, TLRs may represent a double-edged sword which may elicit dysregulated immune signaling, leading to the production of proinflammatory mediators, resulting in severe disease. TLR-dependent excessive inflammation and TLR-dependent antiviral response may tip the balance towards the former or the latter, altering the equilibrium that drives the severity of disease.

## 1. Introduction

The damaging effects of pathogens have led to the evolution of a variety of host-defense mechanisms, among which innate immune responses are of prime importance in the early recognition and elimination of invading microorganisms, including severe acute respiratory syndrome coronavirus-2 (SARS-CoV-2) [1].

SARS-CoV-2, a positive-sense single-stranded (ss)RNA Betacoronavirus member of the family *Coronaviridae*, emerged at the end 2019 and is the causative viral pathogen of the coronavirus disease 2019 (COVID-19), which became pandemic in March 2020. Infection with SARS-CoV-2 causes mild to moderate respiratory illness in the majority of patients; however, it may occasionally be responsible for severe interstitial pneumonia, myocarditis, acute kidney injury, acute respiratory distress syndrome (ARDS), multiorgan failure and death, predominantly in elderly patients and in those with several comorbidities such as cardiovascular disease, diabetes, chronic respiratory disease, or cancer or genetic predisposition [2,3,4,5] Clearly, beyond SARS-CoV-2 variability that contributes to incomplete immune protection, inflammatory conditions, as well as the immune status of patients, are critical in determining the course of the disease [6,7].

Innate immune recognition is based primarily on complement activation, phagocytosis, autophagy and immune activation by a limited number of germline-encoded pattern recognition receptors (PRRs) that recognize conserved motifs on pathogens named pathogen associated molecular patterns (PAMPs), whose individual variability is genetically determined. Importantly, PRRs are expressed constitutively in the host, and sense the pathogens regardless of their life-cycle stage [8,9]. PRRs are present in both immune and non-immune cells, and induce the release of a selected repertoire of host defense factors, inducing a specific inflammatory response driven by the pathogen’s molecular signatures. In mammalian cells, PRRs can be classified into two main classes: membrane-bound receptors, such as Toll-like receptors (TLRs) and C-type lectin receptors (CLRs), and cytoplasmic sensors, including NOD-like receptors (NLRs), absent in melanoma-2-like receptors (ALRs), RIG-I-like receptors (RLRs) and an increasing range of cytosolic nucleic acid sensors [8,9] (Figure 1).

## 2. Structure and Downstream Signaling Pathway of TLRs

The TLR family is one of the best-characterized PRR families, and is responsible for sensing invading pathogens outside the cell and in intracellular endosomes and lysosomes, leading to a potent immunostimulatory response. In humans, 10 TLRs respond to a variety of PAMPs, including lipopolysaccharide (TLR4), lipopeptides (TLR1, TLR2, TLR6 and TLR10), bacterial flagellin (TLR5), viral dsRNA (TLR3), viral or bacterial ssRNA (TLR7 and TLR8) and CpG-rich unmethylated DNA (TLR9). TLR1, TLR2, TLR4, TLR5, TLR6 and TLR10 are plasma membrane TLRs, whereas TLR3, TLR7, TLR8 and TLR9 are located on endosomes [11,12,13,14,15,16] (Figure 1). TLRs are type I integral membrane glycoproteins, characterized by extracellular (or extraendosomal) domains containing varying numbers of leucine-rich-repeat (LRR) motifs and a cytoplasmic signaling domain homologous to that of the interleukin 1 receptor (IL-1R), termed the Toll/IL-1R homology (TIR) domain. TLRs share a common structural framework in their extracellular, ligand-binding domains. These domains all adopt horseshoe-shaped structures formed by LRR motifs. Classically, in ligand binding, two extracellular domains form an “m”-shaped dimer, sandwiching the ligand molecule and bringing the transmembrane and cytoplasmic domains into close proximity and triggering a downstream signaling cascade. Although the ligand-induced dimerization of these receptors has many common features, the nature and number of the interactions of the TLR extracellular domains with their ligands varies markedly between TLRs [17]. For instance, crystallographic studies of TLR7 and TLR8 have revealed that these receptors contain two distinct ligand-binding sites that cooperate to maximize dimerization. Site 1, highly conserved in both TLRs, recognizes nucleosides (guanosine for TLR7 and uridine for TLR8) and is essential for receptor dimerization and signaling with ad hoc ligand concentration. The other binding site (Site 2) recognizes nucleic acid sequences and plays an auxiliary role in receptor dimerization by enhancing the binding affinities of site 1 ligands. This process, by which dual ligands (nucleic acid sequences and free nucleosides) concur to maximally dimerize TLRs, takes place only in the endosome. Of note, ssRNA-binding to site 2 is not sufficient for the formation of a signaling competent TLR dimer, highlighting the relevance of these TLRs to sense distinct RNA-degradation products rather than full-length ssRNAs [18,19,20,21,22]. Extracellular domains of TLRs can be homodimers, in the cases of TLRs 3, 4, 5, 7, 8 and 9, or heterodimers as in TLRs 1 and 2 or TLR2 and 6 [14,23,24] (Figure 1).

After ligand-induced dimerization, the cytoplasmic TIR domain associates with TIR domain-containing adaptor molecules to transmit signaling. To date, six adaptor proteins have been identified: MyD88 (myeloid differentiation primary-response gene 88), MAL (MyD88-adaptor-like protein), TRIF (TIR-domain-containing adaptor protein inducing interferon-β (IFNβ)) and TRAM (TRIF-related adaptor molecule) are recruited to TIR domains to initiate signaling, whereas SARM (sterile α- and armadillo-motif-containing protein) and BCAP (B-cell adaptor for PI3K) inhibit TLR responses. All TLRs, except for TLR3, associate with MyD88 and MAL proteins. Otherwise, TLR3 and TLR4 use a TRIF-dependent pathway. Both the MyD88-dependent and TRIF-dependent pathways lead to the activation of downstream molecules: nuclear factor-κB (NF-κB), activating protein-1 (AP-1), and members of the IFN-regulatory factor (IRF) family [17,25,26,27,28].

The transcription factor NF-κB is the master regulator of all TLR-induced responses. There are seven NF-κB proteins (p65, c-Rel, RelB, p105, p50, p100 and p52), all of which share the Rel homology domain (RHD) that binds to discrete DNA sequences known as κB sites present in promoter and enhancer regions of various genes. The most frequently activated form of NF-kB in TLR signaling is a heterodimer composed of RelA and p50. The transcriptional activity of NF-κB is regulated by the inhibitory kappaB (IκB) proteins that prevent translocation to the nucleus where NF-κB binds their cognate sites in DNA and activates gene transcription. This pathway, called the ‘canonical pathway’, is responsible for the TLR-mediated induction of inflammatory cytokines such as tumor necrosis factor-α (TNF-α) and IL-6 [29] (Figure 2).

## 3. TLRs and SARS-CoV-2

Several TLRs, such as TLR2, TLR3, TLR4 and TLR7, have been associated with COVID-19 severity. Here, we review the roles of TLRs in the pathogenesis of COVID-19, including TLR7 and TLR3 rare variants, as well as the L412F polymorphism in TLR3 that negatively regulates anti-SARS-CoV-2 immune responses, the interaction of TLR2 and TLR4 with SARS-CoV-2 proteins, TLR3-related cellular senescence and the implication of TLR2 in NET formation by SARS-CoV-2 (Figure 3).

### 3.1. TLR7

The constitutive expression of TLR7 is predominant in plasmacytoid and myeloid dendritic cells (pDCs and mDCs), B cells and monocytes, as compared with Natural Killer (NK) cells and T cells, which showed only marginal levels of TLR7 [39,40,41]. Low levels of TLR7 have also been observed in non-immune cells such as hepatocytes and epithelial cells, as well as nasal epithelial cells and keratinocytes [42,43,44,45,46].

A bioinformatic analysis showed that the related viruses SARS-CoV-1 and Middle East Respiratory Syndrome Coronavirus (MERS-CoV) possess a high number of binding motifs for TLR7, and that the SARS-CoV-2 genome has even more ssRNA motifs that can potentially interact with TLR7 than its family member SARS-CoV-1 [47].

Van der Made et al. [48] were the first to describe deleterious variants in the X-chromosomal TLR7 gene associated with the pathogenesis of COVID-19. They reported two families with affected males at a mean age of 26 years, with no history of major chronic disease, characterized by rare loss-of-function (LOF) TLR7 variants who suffered from severe COVID-19. Whole-exome sequencing identified both a maternally inherited 4-nucleotide deletion p.(Gln710Argfs*18) and a missense mutation p.(Val795Phe). Stimulation with the TLR7 agonist imiquimod (IMQ) of primary peripheral blood mononuclear cells (PBMC) from the patients resulted in a transcriptionally impaired host type I IFN response downstream of the TLR7 pathway, as evidenced by the impaired upregulation of IRF7, IFNβ1 and interferon-stimulated gene ISG15, and by an abrogated production of IFNγ, supporting the importance of intact TLR7 signaling in COVID-19 pathogenesis. After this sporadic observation, the first paper [49] statistically demonstrating the relevance of TLR7 came from the Italian consortium GEN-COVID (accessed on 16 March 2020, https://sites.google.com/dbm.unisi.it/gen-covid). The relevance of TLR7 was then confirmed by several other studies [39,50,51]. In the Italian cohort study, eight male cases, six aged less than 60 years and two in their mid-60s, with severe COVID-19 carried three rare missense LOF [p.(Ser301Pro), p.(Arg920Lys) and p.(Ala1032Thr)], one hypomorphic p.(Val219Ile) and two less functionally impacting variants [p.(Ala288Val) and p.(Ala448Val)] [47]. Therefore, a new entity of “X-linked Mendelian disorder conditioned by viral infection” was definitively established. The disease segregates in families though females and mutated males are affected.

In a further follow-up study with an enlarged GEN-COVID cohort, Mantovani et al. reported another variant, p.(Asp41Glu), predicted to be deleterious by in silico analysis in a 79-year-old severely affected and deceased male patient, whose not-yet-infected mutated brothers were highly at risk of severe disease if not permanently protected by vaccine. These studies demonstrated an impaired upregulation of IRF7, ISG15, CXCL10, RSAD2, ACOD1, IFIT2, IFNα and IFNγ genes, highlighting a profound impairment of the TLR7 signaling pathway in response to the TLR7 agonist [51]. Solanich et al. [50] identified two additional, previously unreported LOF variants in a cohort of young males affected with severe COVID-19. One, p.(Asn215Ser), was detected in a 30-year-old male patient and the other p.(Trp933Arg) in a 28-year-old male patient. Functional testing of the p.(Trp933Arg) variant revealed decreased type I and II IFN production, similar to previously published findings. Fifteen very rare TLR7 variants were reported in a larger cohort of 1202 male patients aged 0.5 to 99 years (mean: 52.9 years) with unexplained critical COVID-19 pneumonia. In a total of sixteen unrelated male individuals aged 7 to 71 years (mean, 36.7 years), twelve had LOF and three had hypomorphic variants [39]. One variant, p.(Leu988Ser), was recurrently found in three patients, including one patient carrying two very rare variants: p.(Met854Ile) and p.(Leu988Ser). Importantly, this study showed that the patients’ peripheral blood pDCs responded poorly to SARS-CoV-2 infection. Indeed, IFNα production by TLR7-deficient pDCs was impaired but not entirely abolished upon SARS-CoV-2 infection. Thus, SARS-CoV-2 triggered type I IFN induction in pDCs in a manner that was at least partially dependent on TLR7.

A de novo hemizygous deleterious mutation in the TLR7 gene, p.(Leu372Met), was reported in a 7-year-old male patient who developed critical COVID-19 pneumonia with preexisting hyper IgM syndrome and Ataxia-Telangiectasia (A-T) [52]. It is unknown if the antibody and T cell defects associated with the ATM deficiency may contribute to severe COVID-19. Although severe acute infections are uncommon [53], viral infections have been reported in 25–30% of A-T patients, mainly in the respiratory tract, predominantly caused by usually asymptomatic Rhinovirus infection [54,55]. Of 247 A-T patients evaluated, 36 had SARS-CoV-2 infection, all with mild or no symptoms, except for the patient carrying the TLR7 variant who had had a critical clinical presentation and was transferred to the intensive care unit (ICU), suggesting that ATM mutation is probably not critical for COVID-19 development [52]. This combined TLR7-ATM deficient case illustrates a potential synergistic impact of defective type I IFN and humoral immune responses which deserves further investigation and should be considered in future studies.

Single-nucleotide polymorphisms (SNPs) of the TLR7 gene have been associated with susceptibility to different infectious diseases, such as HIV-1 and HCV, and with respiratory diseases [56,57]. Among these, the frequent SNP of the TLR7 coding sequence, rs179008 (NM_016562.3:c.32A>T), was associated with higher viral loads and accelerated progression to advanced immune suppression in HIV positive male patients, and with a diminished cytokine response to a TLR7 agonist in leukocytes of male allele T carriers [58]. Azar P et al. [59] found that carriage of the rs179008-T allele in HIV-positive females was associated with both a decrease in TLR7 protein relative to A/A homozygous pDCs and lower per-cell IFN-I production. The TLR7 rs179008A/T SNP also modulates the clearance and progression of HCV infection with different magnitudes between sexes, with the T allele increasing the risk of disease progression in both sexes [60]. TLR7 rs179008 polymorphism showed a strong association with the development of bronchial asthma, with the most significant association in boys, suggesting a predominantly recessive effect of these variants [57]. The minor allele rs179008-T is common worldwide, except in East Asia, and is especially frequent among European populations, where 30–50% of women are homozygous or heterozygous carriers. The rs179008 is a missense SNP substituting a leucine for a glutamine (p.Gln11Leu) at protein level, associated with decreased TLR7 protein synthesis and homodimer formation in vitro [59]. To date, few studies have investigated the association of rs179008 SNP with SARS-CoV-2 infection and COVID-19 development. One study found statistically non-significant differences between patient groups for the TLR7 rs179008 A/T allele in mild, moderate and severe COVID-19 [61]. In contrast, in another study the mutant T allele was associated with an increased risk of COVID-19 pneumonia but not with disease outcome [62]. Moreover, SARS-CoV-2 was associated with hepatitis in a male child carrying the rs179008-T allele, suggesting that hepatitis purportedly caused by SARS-CoV-2 infection could be associated with inefficient initial innate immune responses against the virus caused by the polymorphism [63]. Another TLR7 SNP (rs3853839G/C) had a relationship with Dengue virus infection [64], with HIV-1 infection and prognosis [65] and with Chikungunya virus infection in Indians [66], and was linked in Chinese patients to HCV persistence and predisposition to enterovirus-71-mediated hand, foot and mouth infection [67,68]. The function of rs3853839 and the expression of the TLR7 mRNA transcript in the development, severity and progression of COVID-19 was investigated in a study which suggested that the G/G genotype and the G allele could be a genetic risk factor for COVID-19 development, severe illness and poor clinical outcome in middle-aged individuals without comorbidities. Furthermore, patients with the GG genotype had the highest levels of TLR7 mRNA expression, whereas those with the CC genotype had the lowest [69]. In conclusion, despite several findings indicating that these SNPs are involved in susceptibility to a broad range of infectious diseases and inflammatory phenotypes, few, not-gender-related data were reported in relation to SARS-CoV-2 infection and COVID-19 development, highlighting the need for in-depth studies.

A male sex bias has emerged in the COVID-19 pandemic, fitting to the sex-biased pattern in other viral infections. A meta-analysis of 3,111,714 globally reported cases demonstrated that, whilst there was no difference in the proportion of males and females with confirmed COVID-19, male patients had an almost three times higher probability of requiring ICU admission and death compared to females. With a few exceptions, the sex bias observed in COVID-19 was a worldwide phenomenon [70,71,72,73]. The X-linked TLR7 gene escapes X chromosome inactivation in 30% of female immune cells, including pDCs, and was associated with higher TLR7 protein expression in leukocytes from rs179008 A/A women compared with A/0 men [74]. As a consequence, a sex bias was reported in the TLR7-driven production of IFNα by human pDCs, with higher frequencies of IFNα–producing cells in adult females than in adult males [59,75,76]. The sex bias in the TLR7-mediated response of pDCs arises from independent mechanisms implicating estrogen signaling [77] and the X-chromosome complement [78]. The X chromosome inactivation escape of TLR7 and the subsequent increased expression of type I IFN may explain in part the observed sex differences concerning severe COVID-19 susceptibility. Severe COVID-19 has been associated with a reduction in circulating pDCs, as well as a minimal influx of pDCs into the lungs compared with patients with moderate disease and healthy controls, and with an impaired type I IFN response in patients with severe SARS-CoV-2 infection [79,80,81,82,83,84,85]. An increase in TLR7-mediated IFN production in females may prevent females from progressing to severe disease. In the case of SARS-CoV-2, in which initially there was no or only mild cross-reactive pre-existing adaptive immune memory, the success of an early antiviral response mediated by IFNα may be a major determinant of disease outcome.

Overall, X-linked recessive TLR7 deficiency was pathogenic in patients by impairing the production of large amounts of type I IFNs by pDCs. PDCs, while not supporting SARS-CoV-2 replication, are recruited to the infection sites where they can phagocytose infected cells and, in response, produce multiple antiviral and inflammatory cytokines that protect epithelial cells from de novo SARS-CoV-2 infection. TLR7-MyD88 signaling was identified as crucial for the production of IFNs in SARS-CoV-2 infection by the targeted deletion of virus-recognition innate immune pathways [79]. Moreover, the evidence that numerous SARS-CoV-2 proteins interfere with the RIG-I pathway, the other major pathway for the activation of antiviral type I IFN response to infection with RNA viruses, suggests dependence on strong signaling via the TLR7-pathway in the early phase of viral infection. Indeed, by acting on various molecules in the RIG-I-MAVS signaling pathway, SARS-CoV-2 proteins can evade host antiviral responses and promote viral replication [86,87,88]. In this way, dependence on functional TLR7-signaling for initial innate immune responses highlights the crucial role of the X-linked recessive TLR7 deficiency in the pathogenesis of COVID-19. The clinical observations of deleterious TLR7 variants and a poor type I IFN response found in 1–2% across male cohorts of severe cases identify TLR7 as an important PRR in the immune response against COVID-19, especially in younger patients. Therefore, genetic screening for TLR7 primary immunodeficiency may be recommended in young males with severe COVID-19 in the absence of other relevant risk factors. On the other hand, a study from a cohort comprising 5085 participants with severe COVID-19 and 571,737 controls that combined whole-exome and whole-genome sequencing and a rare variant burden test meta-analysis observed that carrying a rare deleterious variant in TLR7 was associated with a 5.3-fold increase in severe disease. Importantly, this observation was consistent across both sexes, suggesting that TLR7-mediated genetic predisposition to severe COVID-19 may be a dominant or co-dominant trait, an observation that cannot be made in cohorts limited to male participants [89].

### 3.2. TLR3

Toll-like receptor 3 is broadly expressed in humans; indeed, its expression and functionality have been confirmed in immune and non-immune cells such as mDCs [90], effector CD8 T cells [91], NK cells [92], intestinal epithelial cells [93], lung and dermal fibroblasts [94,95] and different CNS-resident cells [95,96,97,98,99]. It was shown that the transcript levels of TLR3 were overexpressed in the nasopharyngeal epithelial cells of COVID-19 patients with clinical symptoms compared with controls [100], and that a lower peripheral blood TLR3 expression was associated with an unfavorable outcome in severe COVID-19 patients [101]. Several lines of evidence support the idea that TLR-mediated signaling pathways, including the TLR3 one, originate in the cell membrane in various cells (reviewed in [102]), and that cell membrane TLR3 contributes to endosomal TLR-mediated inflammatory signaling pathways. Surface expression was demonstrated on human lung epithelial cells [103] and on several cell lines [102]. To our knowledge, there was no evidence of cell membrane TLR3-mediated responses contributing to responses to SARS-CoV-2.

A role for the TLR3 receptor has been described in host defense against numerous viruses such as herpes simplex virus, hepatitis C and B virus, A/H1N1/2009 influenza virus, rotavirus, tick-borne encephalitis virus and HIV-1 (reviewed in [104]). An in silico study on the binding of all SARS-CoV-2 mRNAs with the intracellular TLRs depicts the NSP10 protein of SARS-CoV-2 being a putative PAMP for TLR3, triggering downstream cascade reactions [105].

Bortolotti et al. [106] showed in an in vitro system in which SARS-CoV-2 infection of Calu-3/MRC-5 multicellular spheroids induced the activation of the TLR3 pathway via IRF3, leading to pro-inflammatory cytokine secretion, including IL-1α, IL-1β, IL-4, IL-6, IFNα and IFNβ, during the first 24 h post-infection.

The first evidence of the involvement of TLR3 and type I IFN immunity in the control of SARS-CoV-2 infection was in the early phases of the pandemic, when four LOF variants [p.(Ser339fs/WT), p.(Pro554Ser/WT), p.(Trp769Ter/WT) and p.(Met870Val/WT)] were reported in four patients with life-threatening COVID-19 pneumonia aged 44 to 77 years [107]. Of note, the same p.(Pro554Ser) variant has been previously reported in patients with life-threatening influenza pneumonia [99,108]. IFNL1 mRNA levels were impaired in TLR3-deficient P2.1 fibrosarcoma cells stably transfected with plasmids expressing WT or mutant forms of TLR3 identified in patients. Importantly, serum levels of IFNα in the patient carrying the p.(Pro554Ser/WT) variant during the acute phase of COVID-19 were considerably lower than those found in the cohort of patients hospitalized with unexplained severe COVID-19.

A single subcutaneous injection of Peg-IFNα2 was administered to a 25-year-old woman with known TLR3 deficiency [p.(Pro455Ser)] and COVID-19 who had two episodes of herpes simplex encephalitis in childhood, but had no further serious viral infections. Thirty-six hours after administration, the viral load had strongly decreased and the patient reported a partial resolution of anosmia within the next 48 h. Although these findings cannot be considered a formal demonstration of the efficacy of IFNα2a therapy for COVID-19, the case report showed that a single subcutaneous injection of Peg-IFNα2a was safe and effective in the patient with impaired production of type I IFNs associated with a predisposition to severe COVID-19 [109].

In a later study, Povysil et al. [110], looking at a substantially larger cohort of 713 patients with severe COVID symptoms, 1151 with mild disease, and a control population of 15,033, found a rare predicted LOF TLR3 variant [p.(Arg867*)] in a mild case and three other variants in controls. However, they found no evidence of the enrichment of rare, protein-altering variants in the TLR3 gene in severe cases. However, it is important to emphasize that in the two studies there were differences in the ancestry and mean age of cohorts, demonstrating how important it is to adequately control for differences between cases and controls, as the frequency of rare variants strongly depends upon ancestry, with Europeans showing higher rates compared with Middle Eastern individuals. Another study identified in the Chinese population a TLR3 variant, the p.(Arg394Ter), in a patient with severe disease [111]. Although they have not undertaken any functional studies, the variant was predicted to be an LOF.

Among TLR3 variants, the rs3775291 L412F polymorphism p.(Leu412Phe) is known to decrease TLR3 expression on the cell surface [112]. This polymorphism leads to poor recognition of SARS-CoV-2 dsRNA during replication, compared to its WT allele [113], and it has been associated with susceptibility to and mortality from SARS-CoV-2 [114]. The L412F polymorphism was also identified as a marker of severity in COVID-19, further demonstrating an increased association in the sub-cohort of males [115]. Indeed, a functional role for the L412F polymorphism has been demonstrated in males with severe COVID-19. Interestingly, the mutation specifically affected TLR3-dependent autophagy and reduced TNF-α production after stimulation of the HEK293 cells transfected with L412F-encoding plasmid with a TLR3 specific agonist. Moreover, a reduced survival rate at 28 days was shown in L412F COVID-19 patients treated with the autophagy-inhibitor hydroxychloroquine, suggesting that the outcome of clinical trials with hydroxychloroquine should be reinterpreted in the light of the TLR3-L412F polymorphism status [116].

Recently, a case–control study reported an association of the mutant T/T genotype of TLR3 (rs3775290) with an increased risk of developing COVID-19 pneumonia independently of disease outcome [62].

The clinical severity of COVID-19 is largely determined by host factors. Recent advances point to cellular senescence, an aging-related switch in cellular state, as a critical regulator of SARS-CoV-2-induced hyperinflammation. SARS-CoV-2, like other viruses, can induce senescence and exacerbates the senescence-associated secretory phenotype (SASP), which is composed largely of pro-inflammatory, extracellular matrix-degrading, complement-activating and pro-coagulatory factors secreted by senescent cells [116]. It has been reported that the surface protein S1 of SARS-CoV-2 induces senescence in human non-senescent cells. Of note, SARS-CoV-2 exacerbates the tissue-destructive SAPS, increasing IL-1α, IL-1β, IL-6, IL-8 and Granulocyte-Macrophage Colony-Stimulating Factor (GM-CSF) mRNA levels in human senescent cells through TLR3. The genetic or pharmacological inhibition of TLR3 prevented senescence induction and SASP amplification by SARS-CoV-2, suggesting that the induction of cellular senescence and SASP amplification through TLR3 could contribute to SARS-CoV-2 morbidity, suggesting that clinical trials with senolytics and/or SASP/TLR3 inhibitors for alleviating acute long-term SARS-CoV-2 sequelae are eagerly warranted [117].

### 3.3. TLR2

TLR2 is a cell surface sensor expressed in immune cells, endothelial, and epithelial cells, including respiratory epithelial cells [118].

TLR2 expression was associated with COVID-19 severity, being identified as a sensor of SARS-CoV-2 that drives inflammatory cytokine production, potentially contributing to the dysregulated immune response observed in patients with severe COVID-19 [119]. SARS-CoV-2 E protein, but not S or M proteins, interacted directly with TLR2, inducing inflammatory responses and the secretion of TNF-α, IFNγ, IL-1α, IL-6, CXCL10, MCP-1, G-CSF and CCL3 in human PBMC. Of note, some of these cytokines have emerged early as critical parameters in COVID-19 disease progression and predictors of disease severity and death [120,121]. Moreover, elegant experiments using heat-inactivated viruses to block transcription and entry demonstrated that the TLR2-mediated activation of inflammatory signaling was independent of viral entry and replication [119]. In vivo infection models demonstrated that the E protein induced TLR2-dependent inflammation in the lungs, and that blocking TLR2 signaling provided protection against SARS-CoV-2 induced pathology [119]. Another study has identified the S protein, and not the E protein, as a driver of TLR2 activation and subsequent cytokine production in the SARS-CoV-2 infection of human and mouse macrophages [122].

Multiple studies have shown that elevated levels of IL-6 in COVID-19 patients were associated with disease severity [80,123,124,125]. Interestingly, it has been demonstrated that pDC, while not directly supporting SARS-CoV-2 replication, senses the virus via TLR2, and secretes the inflammatory cytokine IL-6, further highlighting how TLR2 responses are critically involved in the cytokine storm elicited by SARS-CoV-2 infection [79].

COVID-19 also impacts the cardiovascular system, resulting in myocardial damage, and affects the kidneys, leading to renal dysfunction [126]. In a small cohort of 50 healthy control and 100 COVID-19 patients with severe or moderate disease, the expression levels for TLR2 and TLR4 mRNA were positively correlated with kidney and heart function biomarkers in serum [127], suggesting that the TLR2 and TLR4 sensors are possible targets to prevent the severity of COVID-19.

Excessive neutrophil extracellular traps contributed to immunothrombosis, leading to extensive intravascular coagulopathy and multiple organ dysfunction in COVID-19 patients [128]. Sung and colleagues [129] demonstrated that SARS-CoV-2-activated platelets produced extracellular vesicles that enhanced thrombo-inflammation via the C-type lectin member 5A and TLR2, suggesting that TLR2 may represent a potential therapeutic target to reduce the risk of acute respiratory distress syndrome in COVID-19 patients.

The TLR2 sensor was considered a potential target to support vaccination. Indeed, the TLR2 ligand, Pam_2_Cys, a highly effective mucosal adjuvant for peptide or protein-based vaccines [130,131,132,133], was tested with recombinant trimeric SARS-CoV-2 Spike protein in a novel subunit vaccine in mice delivered via the mucosal or systemic route [134]. Both routes of vaccination induced substantial neutralizing antibody titers; however, mucosal vaccination uniquely generated anti-Spike IgA, increased the neutralizing antibody in the serum and airways, and increased lung CD4+ T-cell responses, supporting mucosal vaccination as a strategy to improve protection in the respiratory tract against SARS-CoV-2 [134]. Moreover, in a SARS-CoV-2 ferret infection model, prophylactic intranasal administration of the TLR2/6 agonist INNA-051 effectively reduced levels of viral RNA in the nose and throat, supporting the clinical development of a therapy based on prophylactic TLR2/6 innate immune activation to reduce SARS-CoV-2 transmission and providing protection against COVID-19 [135].

Overall, several findings indicate that TLR2 may be exploited to improve the mucosal vaccination strategy via TLR2 agonists. This may provide better protection for the respiratory tract and, in contrast, TLR2 antagonists may protect against the cytokine storm and disease progression.

### 3.4. TLR4

Innate immune cells, such as circulating monocytes, tissue macrophages and DCs, express TLR4. Analyses of total RNA showed that TLR4 mRNA was present only in myeloid cells and was undetectable in resting or activated lymphoid cell subsets. TLR4 is constitutively expressed in adipocytes, microglia and in the macroglial cells such as astrocytes, dermal microvessel endothelial cells and umbilical vein endothelial cells [136]. Despite TLR4 mainly residing in the plasma membrane, it can also be considered as an intracellular TLR because it can be internalized and stimulate intracellular pathways [137].

Computer-based modeling has predicted the binding of TLR4 to the Spike protein [138,139]. Consistently, it has been reported in murine and human macrophages that Spike protein S1 subunit binds to TLR4, inducing pro-inflammatory responses and the activation of transcription factors such as NF-κB, AP-1 and encoding proinflammatory cytokines and IFNs [140,141,142]. Blocking Spike-TLR4 interaction could therefore be a potential target for regulating excessive inflammatory responses in COVID-19 patients [143].

Endothelial cells (EC) are sentinels uniquely positioned at the interface between circulating blood and underlying tissues that gatekeep micro- and macro-vascular health by sensing pathogen signals and secreting vasoactive molecules. SARS-CoV-2 infection primarily affects the pulmonary system, but accumulating evidence suggests that it also affects the pan-vasculature in the extrapulmonary systems by directly or indirectly generating a cytokine storm, causing endothelial dysfunction such as endotheliitis, endothelialitis and endotheliopathy [144]. To date, the direct causative mechanism of SARS-CoV-2-induced endotheliitis remains unclear. Ma et al. [145] contributed to define a role for TLR4 in the ACE2-independent inflammatory activation of vascular EC. ACE2-deficient ECs respond to SARS-CoV-2 through TLR4, as demonstrated by treatment with its antagonists, CLI095 or dexamethasone, which inhibited p38 MAPK/NF-kB/IL-1β activation after viral exposure. Genome-wide and single-cell RNA-seq analyses further confirmed the activation of the TLR4/MAPK14/RELA/IL-1β axis in circulating ECs in patients with mild and severe COVID-19, indicating that human ECs were not protected from SARS-CoV-2-mediated activation by being ACE2-deficient as they still recognized and responded to the virus through TLR4 activation. Interestingly, dexamethasone has been demonstrated to lower the 28-day mortality rate among patients hospitalized with COVID-19 receiving respiratory support [146], providing a clinical benefit that could also be explained by the inhibition of TLR4-mediated activation in the endothelium.

Neutrophils, among the first line of innate immune defense in acute infection, extravasate rapidly from the blood vessels into tissues to engulf and kill pathogens. Various stimuli, such as microorganisms, pathogens, activated platelets and pro-inflammatory cytokines, can trigger a rapid and active process of cell death in neutrophils, ending up in the formation of neutrophil extracellular traps (NETs) [147]. Numerous pieces of evidence have implicated neutrophils and the imbalance between NET formation and degradation in the pathophysiology of inflammation, coagulopathy, organ damage and immunothrombosis that characterizes severe cases of COVID-19 [148,149]. Moreover, in patients with severe disease, an increased neutrophil-to-lymphocyte ratio, a high expression of the neutrophil-related cytokines IL-8 and IL-6 in serum and elevated levels of circulating calprotectin, a neutrophil-mediated inflammatory protein, have been described as predictors of poor outcome [150,151,152,153,154,155,156,157]. The mechanism controlling calprotectin production during SARS-CoV-2 infection was elucidated by Loh et al. [158]. It has been demonstrated that calprotectin expression in neutrophils is regulated via the TLR4 pathway in response to Spike protein, and that the adaptor Docking Protein 3 restrained the production of calprotectin during infection when TLR4 was engaged.

**Figure 3 ijms-24-08065-f003:**
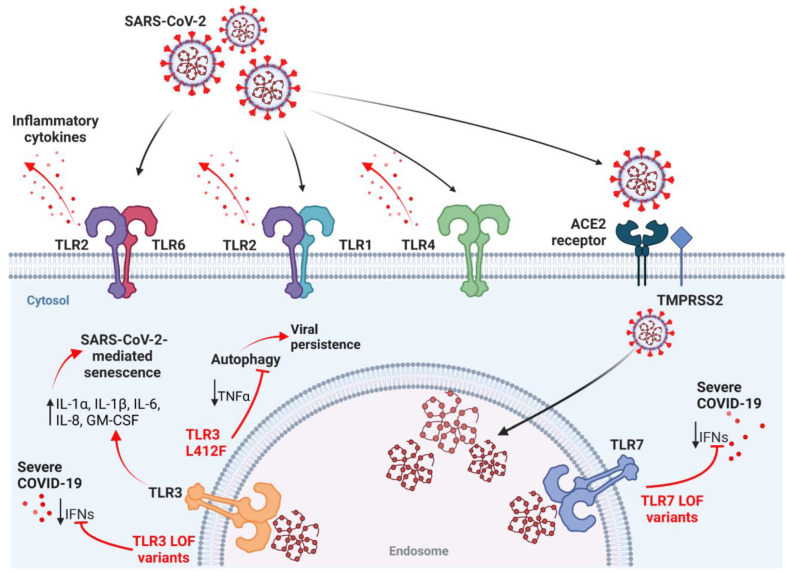
Several TLRs, such as TLR2, TLR3, TLR4 and TLR7, have been associated with the pathogenesis of COVID-19. The interaction of TLR2 and 4 with SARS-CoV-2 proteins triggers downstream pathways leading to the production of pro-inflammatory cytokines, via the activation of multiple adaptor proteins, as shown in Figure 2 [119,122,140,141,142,143]. LOF TLR3 and TLR7 variants negatively regulate anti-SARS-CoV-2 immune responses, inducing an impaired IFN response that could delay viral clearance in the initial disease phase and promote a severe COVID-19 clinical course [39,48,49,50,51,107]. The L412F polymorphism reduces TNFα production and inhibits TLR3-dependent autophagy, promoting viral persistence [115]. SARS-CoV-2 induces senescence in human non-senescent cells and exacerbates the tissue-destructive SAPS, increasing IL-1α, IL-1β, IL-6, IL-8 and GM-CSF mRNA levels in human senescent cells through TLR3 [117]. SAPS, senescence-associated secretory phenotype. Created with BioRender.com.

## 4. TLRs, IL-1 and Cytokine Storm Syndrome

During TLR-mediated responses induced by SARS-CoV-2 infection, the release of pro-inflammatory cytokines contributes to cause the cytokine storm syndrome that is associated with heart and lung tissue damage, leading to multiorgan failure and poor outcomes [122,159,160]. Cytokine storm syndrome results from a sudden acute increase in circulating levels of different pro-inflammatory cytokines, including IL-6, IL-1, TNF-α and IFNs [161]. IL-1β, due to its pro-inflammatory and pleiotropic biological activity, plays a central role in orchestrating the innate immune responses following a wide range of conditions, including infections [162,163]. The interleukin-1 (IL-1) family of cytokines and their receptors share a structure and function similar to TLRs. Cytokine members of the IL-1 family trigger innate inflammation via the IL-1 family of receptors, increasing nonspecific resistance to infection and developing immune responses to foreign antigens. Each member of the IL-1 receptor, as for the TLR family, contains a cytoplasmic portion formed by a TIR signaling domain. Once IL-1 binds to its receptor, a structural change occurs that allows binding to the co-receptor, forming a trimeric complex, in which the TIR domains approximate. The binding of MyD88 to the TIR domains triggers a cascade of kinases that produces a strong pro-inflammatory signal, via the activation of NF-κB, that amplifies that mediated by TLRs [163]. Interestingly, longitudinal analyses of immune responses in 113 patients with moderate or severe COVID-19 revealed distinct profiles that influenced the evolution and severity of COVID-19. Indeed, following an early increase in a variety of cytokines, patients with moderate COVID-19 displayed a progressive reduction in inflammatory cytokines; by contrast, patients with severe COVID-19 maintained these elevated responses throughout the course of the disease. Inflammasome-induced cytokines, such as IL-1β and IL-18, were higher in patients with severe disease than in patients with moderate disease [164].

The cytokine storm initiates a variety of inflammatory signaling pathways via their receptors on immune and tissue cells, which lead to fever, capillary leak syndrome, disseminated intravascular coagulation, acute respiratory distress syndrome, and multi-organ failure, ultimately leading to death in the most severe cases [160]. Interestingly, in COVID-19 patients, pain, including headache, widespread myalgia, and back and neck pain, was linked to inflammation and cytokine storm, especially with the rapid release of the pro-inflammatory cytokines IL-6, IL-1RA, IL-10, IL21 and IL-22 [165,166,167]. Moreover, pro-inflammatory cytokines can act on pain-sensing nociceptors in the neurons of peripheral tissues and in the central nervous system, heightening perceptions of pain and leading to chronic pain [167]. Another critical symptom of COVID-19 is the so-called brain fog, a pathological state represented by cognitive dysfunction and fatigue. Neurologic impairment in COVID-19 might be due to the passage of inflammatory cytokines from the bloodstream through the blood–brain barrier [168,169]. Of note, cerebral hyperinflammation was detected in COVID-19 patients by electroencephalogram [170]. Inflammatory cytokines could be also involved in anosmia, a frequent clinical feature in COVID-19 patients. One study found elevated levels of the proinflammatory cytokine TNF-α in the olfactory epithelium of patients with COVID-19, suggesting that the direct inflammation of the olfactory epithelium could play a role in acute olfactory loss [171]. However, the mechanisms of olfactory dysfunction in COVID-19 are still unclear [172], and there is evidence that isolated anosmia, which is often associated with mildly symptomatic disease, is associated with robust SARS-CoV-2 specific CD4+ and CD8+ T-cell responses [173]. Anosmia is a prominent clinical characteristic of Parkinson’s disease (PD); of note, SARS-CoV-2 infection has been predicted as a potential risk factor for developing Parkinsonism-related symptoms in COVID-19 patients [174]. A study reported that two of four patients with long-term olfactory dysfunction, still persisting 4–9 months after SARS-CoV-2 infection, showed damage in the basal ganglia after infection, highlighting the presence of an olfactory-nigral dysfunction, possibly representing a pre-clinical signature of Parkinsonism [175]. Further studies are needed to address the role of inflammatory mediators, activated by SARS-CoV-2 infection, in neurological disorders.

## 5. Conclusions

Is TLR activation beneficial or harmful in SARS-CoV-2 infection?

An early and strong protective TLR-mediated innate immune response against pathogenic viruses or viral components represents an essential contribution to viral clearance. Indeed, TLR signaling is needed for the secretion of antiviral cytokines, chemokines and IFNs, especially type I IFNs, which help to control infection and to determine the disease outcome. If TLR-mediated responses are deleted or reduced, they can lead to severe disease and fatal outcomes, due to the inability to mount effective innate immune responses. In COVID-19, a peculiar phenotype was observed in critical disease, consisting of a highly impaired IFN type I response (characterized by no IFNβ and low IFNα production and activity) and a low ISG signature in peripheral blood, which was associated with a persistent blood viral load and an exacerbated inflammatory response [80]. The dependence on functional TLR7-signaling for initial innate immune responses was highlighted in male cohorts of severe cases, especially in younger patients. Moreover, SARS-CoV-2 made use of evasion mechanisms to protect against viral dsRNA and ssRNA recognition by TLRs.

On the other hand, TLRs may represent a double-edge sword which may elicit dysregulated immune signaling. Indeed, excessive TLR activation due to overstimulation by viral proteins or by damage-associated molecular patterns released from lysed cells can lead to the untoward production of proinflammatory cytokines and chemokines, resulting in severe disease. Hadjadj J. et al. [80] demonstrated that in COVID-19, besides an impaired type I IFN response, severe disease correlates with an immune signature characterized by an early cytokine- and chemokine-rich inflammatory response, partially driven by NF-κB and characterized by increased TNF-α and IL-6 production and signaling, supporting a role for TLR-mediated responses. Indeed, the SARS-CoV-2 E protein interacts directly with TLR2, inducing inflammatory responses and the secretion of a variety of inflammatory cytokines such as TNF-α and IL-6 [122]. TLR4 recognizes the S protein, inducing pro-inflammatory responses and the activation of NF-κB and encoding proinflammatory cytokines [140,141,142]. TLR-dependent excessive inflammation and TLR-dependent antiviral responses may tip the balance towards the former or the latter, altering the equilibrium that drives the severity of disease.

SARS-CoV-2 mRNA vaccines have changed the worldwide perspective on the COVID-19 pandemic. BNT162b2 and mRNA-1273 vaccines maintain the potential to be recognized by intracellular RNA sensors, such as TLR3 and TLR7, and may on the one hand boost mRNA vaccine-induced adaptive immune responses and, on the other, elicit exaggerated innate immunity. Notably, to avoid the latter, mRNA COVID-19 vaccines contain purified, in vitro-transcribed single-stranded mRNA with modified nucleotides to reduce binding to TLR and immune sensors, thus limiting the excessive production of type I IFNs and their inhibitory function on cellular translation [176,177]. A recent study longitudinally profiled the B cell response to mRNA vaccination in SARS-CoV-2 naive patients with inherited type I IFN deficiency, such as a 36-yr-old male patient hemizygous for an LOF TLR7 mutation, demonstrating that the RBD-specific memory B cell response in all patients was quantitatively and qualitatively similar to that in healthy donors [178]. Despite the study being carried out on a patient, these findings strengthen the role for mRNA vaccines to protect patients with inherited deficiencies in type I IFN immunity.

## Figures and Tables

**Figure 1 ijms-24-08065-f001:**
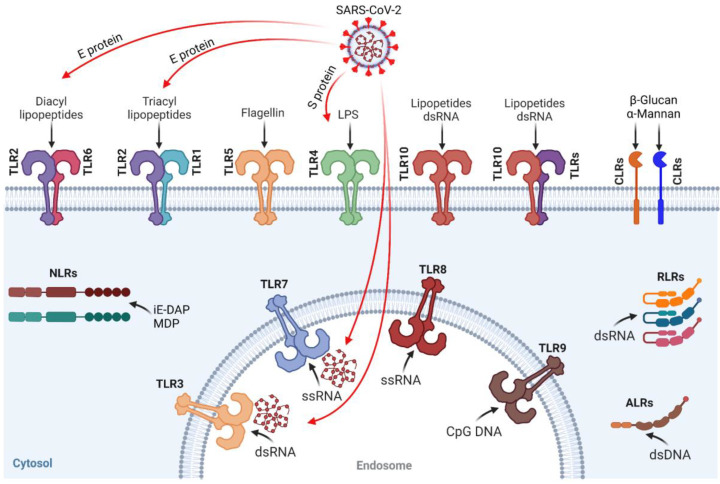
Pattern recognition receptors (PRRs) and their ligands. PRRs can be classified into two main classes: membrane-bound receptors, such as TLRs and CLRs, and cytoplasmic sensors, including NLRs, ALRs and RLRs. In humans, 10 TLRs respond to a variety of Pathogen Associated Molecular Patterns, including lipopolysaccharide (TLR4), lipopeptides (TLR1, 2, 6 and 10), bacterial flagellin (TLR5), viral dsRNA (TLR3 and 10), viral or bacterial ssRNA (TLR7 and 8) and CpG-rich unmethylated DNA (TLR9). TLR1, 2, 4, 5, 6 and 10 are plasma membrane TLRs, whereas TLR3, 7, 8 and 9 are located on endosomes. TLRs can be homodimers, in the cases of TLRs 3, 4, 5, 7, 8 and 9, or heterodimers as in TLRs 1 and 2 or TLR2 and 6. TLR2 and 4 sense the SARS-CoV-2 Envelope protein (E) and the Spike protein (S), respectively, whereas TLR3 and 7 sense SARS-CoV-2 nucleic acid. CLRs are a family of receptors that recognize carbohydrates, such as β-Glucan and α-Mannan, on the surface of pathogenic microorganisms. NLRs are intracellular PRRs that mainly recognize the diaminopimelic acid iE-DAP of the cell wall of Gram-negative bacteria and MDP in all bacterial cell walls. ALRs recognize the dsDNA of bacteria. RLRs recognize short dsRNA (<1000 bp) and long-chain dsRNA (>1000 bp) of different viruses through ligand-recognition domains. TLRs, Toll-like receptors; CLRs, C-type lectin receptors; NLRs, NOD-like receptors; ALRs, absent in melanoma-2-like receptors; RLRs, RIG-I-like receptors; dsRNA, double-stranded RNA; ssRNA, single-stranded RNA; dsDNA, double-stranded DNA; LPS, lipopolysaccharides; CpG DNA, cytosine-phosphate-guanine DNA; iE-DAP, γ-D-glu-meso-diaminopimelic acid; and MDP, muramyl dipeptide. Created with BioRender.comTLRs are crucial components in the initiation of innate immune responses to a variety of pathogens, triggering the downstream production of pro-inflammatory cytokines, interferons (IFNs), and other mediators [10]. Several findings have highlighted the potential contribution of TLRs to the dysregulated immune response observed in patients with severe COVID-19.

**Figure 2 ijms-24-08065-f002:**
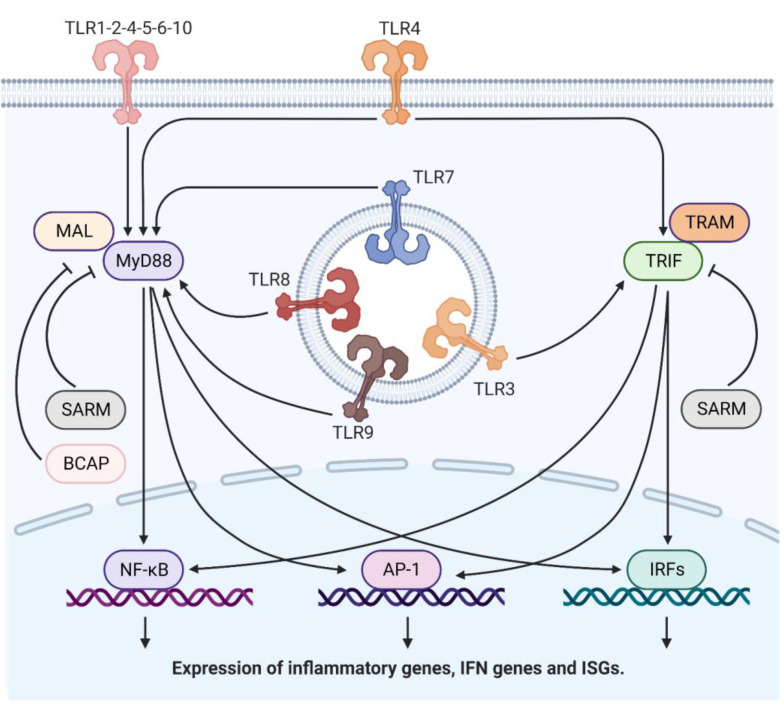
Downstream signaling pathway of TLRs. TLRs are type I integral membrane glycoproteins characterized by the extracellular (or the extraendosomal) domains and a cytoplasmic signaling domain. TLRs share a common structural framework in their extracellular ligand-binding domains. These domains, formed by varying numbers of leucine-rich-repeat motifs, all adopt horseshoe-shaped structures. After ligand binding, two extracellular domains form an “m”-shaped dimer sandwiching the ligand molecule, bringing the transmembrane and cytoplasmic domains into close proximity and triggering a downstream signaling cascade. After ligand-induced dimerization, the cytoplasmic signaling domain associates with adaptor molecules to transmit signaling. Six adaptor proteins have been identified: MyD88, MAL, TRIF and TRAM are recruited to TLR cytoplasmic domains to initiate signaling, whereas SARM and BCAP inhibit TLR responses. All TLRs, except for TLR3, associate with MyD88 and MAL proteins. Otherwise, TLR3 and TLR4 use a TRIF-dependent pathway. Both the MyD88-dependent and TRIF-dependent pathways lead to the activation of downstream molecules: NF-κB, AP-1, and members of the IRF family. These pathways are responsible for the TLR-mediated expression of inflammatory genes, type I, type II and type III IFN genes and interferon stimulated genes (ISGs). MyD-88, myeloid differentiation primary-response gene 88; MAL, MyD88-adaptor-like protein; TRIF, TIR-domain-containing adaptor protein inducing interferon-β (IFNβ); TRAM, TRIF-related adaptor molecule; SARM, sterile α- and armadillo-motif-containing protein; BCAP, B-cell adaptor for PI3K; NF-κB, nuclear factor-κB; AP-1, activating protein-1; and IRF, IFN-regulatory factor. Created with BioRender.com.All nine members of the IRF family have a conserved amino-terminal DNA-binding domain (DBD) [30,31,32,33,34,35] that recognizes the consensus DNA sequence element ISRE [36] in the gene promoters of IFNs and interferon-stimulated gene (ISG) genes [37]. These cytokines, in turn, activate antimicrobial and proinflammatory activities, as well as the maturation of antigen-specific adaptive immune responses. IRF3, IRF5 and IRF7 are primarily responsible for the activation of type I IFN genes downstream of TLR activation, although IRF1 and IRF8 can also contribute [37]. IRF1, IRF3, IRF5 and IRF8 also induce the expression of proinflammatory cytokines such as IL-6, TNF-α, CCL5/RANTES, CXCL10 or CCL2, and other genes in response to TLR activation [38].

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
