# Peer review of "TLRs: Innate Immune Sentries against SARS-CoV-2 Infection"

_ijms, 2023, doi:10.3390/ijms24098065_

Round 1
Reviewer 1 Report
Title: TLRs: innate immune sentries against SARS-CoV-2 Infection.
In this paper, the authors study the coronavirus disease 2019 (COVID-19), caused by severe acute respiratory syndrome coronavirus 2 (SARS-CoV-2). The study deals with the TLRs molecules involved in immunity and inflammation in the COVID-19 disease. The authors conclude that excessive TLR-dependent inflammation and TLR-dependent antiviral response may tip the balance towards the former or the latter by altering the balance that drives disease severity, taking into account that gender and comorbidities may be different. critical variables in unvaccinated patients.
Fig.1, 2, and 3 are very complex and must be explained well in the legend (all steps).
TLR is closely related to IL-1. In this review this part is not mentioned and we talk about pro-inflammatory cytokines. These topics should be briefly included.
- This review paper looks good to me, but it should be better presented. For example it lacks some parts. To make this paper more interesting for the readers of this important journal, the authors, in relation to their data, should talk about cytokines in COVID-19, inflammation and pain and. In this regard, below I report 3 interesting articles that should be studied, incorporate their meaning and report them briefly in the discussion and in the list of references.
M.A. De Rosa, D. Calisi, C. Carrarini, et al. Olfactory dysfunction as a predictor of the future development of parkinsonism in covid-19 patients: a 18f-fdopa pet study. European Journal of Neurodegenerative Diseases 2023; 12(1) January-June (head of print). (www.biolife-publisher.it)
E. Antoniades, S. Melissaris, D. Panagopoulos, E. Kalloniati, G. Sfakianos. Pathophysiology and neuroinflammation in COVID-19. European Journal of Neurodegenerative Diseases 2022; 11(1) January-June: 7-9. (www.biolife-publisher.it)
S.K. Kritas. COVID-19 and pain. European Journal of Neurodegenerative Diseases 2021; 10(2) July-December: 32-35. (www.biolife-publisher.it)
I believe these suggestions are important for improving this paper. Without these corrections the paper cannot be published. So I recommend minor revision.
Reviewer 2 Report
Very interesting and comprehensive review of a current literature on the involvement of different TLRs in SARS-CoV-2 infection and vaccination. Several statements need to be clarified and all figures need to be updated with corresponding information relevant to the main topic of this review.
Specific comments:
1. Abtsract:
a. Unclear statement: TLR2, TLR3, TLR4 and TLR7 have been associated with COVID‐19. Associated with higher severity? With increased viral entry into cells? Clarify the forementioned association.
b. Unclear statement: The activation of TLRs contributes to viral clearance and disease resolution. It is unclear because the activation of TLRs could contribute to disease progression and not all TLRs are activated by virus(es).
c. The reviewer does not see the detailed explanation to “gender …could be critical variables in unvaccinated patients” in the manuscript text. Specify in the text of the manuscript if a gender is not a critical variable in vaccinated patients being infected with SARS-CoV-2 variants.
2. Figures 1 and 2 are incomplete because they depict the already known information about TLRs and individual TLR signaling but do not show how SARS-CoV-2 virus is involved, how it modifies TLR signaling, and how it contributes to infection. Similarly, Figure 3 simply states that TLRs activation by SARS-CoV-2 induces the release of inflammatory cytokines but does not depict by what mechanisms, or how TLR3-dependent autophagy is involved in SARS-CoV-2 infectivity.
3. Conclusions: Clarify how SARS-CoV2 mRNA vaccines have changed the worldwide perspective on the COVID-19 pandemic. What kind of perspectives were in place before vaccines?
Minor comment:
“Severe patients” sounds awkward. Patients with a severe (form of) disease? Severe COVID-19? The reviewer admits seeing a term “severe patients” in other publications.
Round 2
Reviewer 2 Report
No further comments